# Pernicious Attitude of Microbial Biofilms in Agri-Farm Industries: Acquisitions and Challenges of Existing Antibiofilm Approaches

**DOI:** 10.3390/microorganisms10122348

**Published:** 2022-11-28

**Authors:** Sazzad Hossen Toushik, Anamika Roy, Mohaimanul Alam, Umma Habiba Rahman, Nikash Kanti Nath, Shamsun Nahar, Bidyut Matubber, Md Jamal Uddin, Pantu Kumar Roy

**Affiliations:** 1Institute for Smart Farm, Department of Food Hygiene and Safety, Gyeongsang National University, Jinju 52828, Republic of Korea; 2ABEx Bio-Research Center, Azampur, Dakkhinkhan, Dhaka 1230, Bangladesh; 3Department of Biotechnology and Genetic Engineering, Noakhali Science and Technology University, Noakhali 3814, Bangladesh; 4Department of Biotechnology and Genetic Engineering, Mawlana Bhasani Science and Technology University, Tangail 1902, Bangladesh; 5Department of Microbiology and Public Health, Khulna Agricultural University, Khulna 9100, Bangladesh; 6Institute of Marine Industry, Department of Seafood Science and Technology, Gyeongsang National University, Tongyeong 53064, Republic of Korea

**Keywords:** pathogenic biofilm, foodborne pathogen, food safety, antibiofilm control, green approach

## Abstract

Biofilm is a complex matrix made up of extracellular polysaccharides, DNA, and proteins that protect bacteria against physical, chemical, and biological stresses and allow them to survive in harsh environments. Safe and healthy foods are mandatory for saving lives. However, foods can be contaminated by pathogenic microorganisms at any stage from farm to fork. The contaminated foods allow pathogenic microorganisms to form biofilms and convert the foods into stigmatized poison for consumers. Biofilm formation by pathogenic microorganisms in agri-farm industries is still poorly understood and intricate to control. In biofilms, pathogenic bacteria are dwelling in a complex manner and share their genetic and physicochemical properties making them resistant to common antimicrobial agents. Therefore, finding the appropriate antibiofilm approaches is necessary to inhibit and eradicate the mature biofilms from foods and food processing surfaces. Advanced studies have already established several emerging antibiofilm approaches including plant- and microbe-derived biological agents, and they proved their efficacy against a broad-spectrum of foodborne pathogens. This review investigates the pathogenic biofilm-associated problems in agri-farm industries, potential remedies, and finding the solution to overcome the current challenges of antibiofilm approaches.

## 1. Introduction

On earth, nearly 99% of bacterial organisms are likely to dwell in a complex community called a biofilm [1]. In a biofilm, a mixture of communities of mono- and mixed-bacterial species are formed in a heterologous architecture surrounded by extracellular polymeric secretions (EPS) that are comprised of environmental DNA (eDNA), polysaccharides, lipopeptides, and proteins. The secreted EPS matrix helps to defend bacteria against physical, chemical, and biological stresses by improving their capability to retain nutrients and water from their surroundings, which allows them to survive in harsh environmental conditions [2]. The accumulation of a bacterial community and the establishment of biofilms may depend on the locations and the favorable substratum influenced by environmental factors (e.g., temperature, nutrients, community interactions, and osmolarity) [3]. Biofilms are commonly found on moisture-laden surfaces, such as food, conveyer belts, processing instruments, water systems, and packaging lines [4]. Bacterial biofilms are established with the preparation of favorable bases (e.g., biotic and abiotic surfaces) after taking up organic or inorganic molecules (e.g., polysaccharides, glycoproteins, and lipids) [5]. Following the bacterial attachment on surfaces, the community members habitually interconnect by cellular signaling systems, such as quorum-sensing (QS) [6]. The formation and development of bacterial biofilms is a sequential process (Figure 1) consisting of (i) initial attachment, (ii) irreversible attachment and cell-to-cell adhesion, (iii) early development of biofilm proliferation, (iv) maturation, and (v) dispersion [7].

Biofilm formation is a major problem in different agri-farm industries, including dairy, meat, aquatic, and agricultural product processing plants [8,9,10]. It has been estimated that more than 31 known and countless unknown microorganisms are responsible for foodborne illness worldwide and approximately 66% of human diseases are associated solely with pathogenic bacteria [11,12]. The National Institute of Health (NIH) has reported that more than 250 identified and many unidentified diseases are associated with the consumption of unsafe foods, where bacterial biofilms are mostly responsible for ~65% of the microbial and 80% of the chronic infections of humans [13,14]. Spoilage and pathogenic bacteria colonize inside the blending tanks, vats, and piping systems in processing plants; therefore, the formed biofilms threaten the safety and quality of food products. Several factors have also accounted as an influencer on the development of microbial biofilms, in particular foods and food processing plants, such as bacterial strain specificity and their suitable growth conditions (e.g., water, required nutrients, pH, temperature) [15,16]. The majority of previous research to date focuses on planktonic bacteria properties and control. However, biofilm formation by bacteria or other microbes is more resistant (10–10,000 times) to antimicrobial than the planktonic state [13]. They have a barrier to antimicrobial agents that prevent or reduce contact [7,17]. This review aims to outline the pathogenic biofilm-associated problems and their eradication strategies using physical, chemical, and biological antibiofilm agents in the various agri-farm industries.

## 2. Microbial Biofilms in Food Processing Industries

Despite the advantageous behavior of beneficiary microorganisms, the formation of pathogenic biofilms by foodborne microorganisms on food and food processing surfaces could contaminate the raw materials and the processing lines of food products. However, pathogenic contamination that leads to the formation of undesirable biofilms in food industries is still poorly understood and hard to control. Therefore, drawing out the proper outlines of the pathogenic biofilm formation and finding the appropriate remedy to inhibit it in the food industry is crucial.

### 2.1. Biofilm Associated Problems in the Dairy Industry

The presence of foodborne microorganisms in raw products and processing plants, leading to the formation of bacterial biofilms is one of the critical problems facing the dairy industry [10,18]. Following the attachment, bacterial colonies generally persist on the foods and food-processing surfaces, such as processing tanks, vats, and pipelines which continue their involvement in the dairy product contamination and compromise the product quality, economic defeat, and public health safety worldwide [17]. In the dairy processing industry, different classes of microorganisms are involved, based on their advantageous and disadvantageous role of efficacy [7]. A wide variety of thermophilic and cryophilic foodborne microorganisms can occupy and endure in any stage from processing to packaging, due to the inadequate pasteurization procedure and handling of end dairy products [19]. For instance, several pathogenic microbial species (e.g., *Bacillus*, *Citrobacter*, *Enterobacter*, *Pseudomonas*, *Raoultella*, and *Klebsiella* spp.) have been detected from storage tanks and processing pipelines wastewater systems in dairy milk powder processing plants [20]. These acute biofilm formers can persist and induce the formation of bacterial biofilm on food surfaces, leading to taint the milk storage systems by colonization, and spores may remain on the packing surfaces of the end products [17]. Moreover, cryophilic bacteria, such as *Pseudomonas fluorescens*, *P. putrefaciens*, and *Listeria monocytogenes,* could make milk and other dairy products difficult to store since they can thrive at cooling temperatures [21]. Heat-stable lipolytic and proteolytic enzymes produced by pathogenic microorganisms (e.g., *Pseudomonas and Serratia* spp.) have also been accounted as milk and dairy product spoilers, by reducing the product shelf-life and inducing the strong off-flavors, such as bitterness, rancidity, or aged taste [19].

#### Biofilm Control Strategies in the Dairy Industry

Foodborne pathogenic bacteria contamination in dairy raw materials is particularly leading to the formation of biofilms on processing equipment and the surrounding environmental surfaces of dairy plants. Therefore, finding a suitable antimicrobial agent is crucial and the antimicrobial strategies required to consider before applying to the dairy processing plants, include surface chemical modification, surface treatment by means of antimicrobials, manufacturing process optimization, and in-depth knowledge of dairy processing machinery and their cleaning procedures for subsequent bacterial biofilm inhibition [10,21]. In the dairy industry, the standard cleaning practice has an imperative role in controlling foodborne pathogenic bacterial growth and inhibiting biofilm formation in dairy manufacturing equipment [22]. For instance, clean-in-place (CIP) in the dairy plants primarily removes fouling materials and the procedure includes washing milk processing lines with chemicals for cleaning and sanitation, and using more antimicrobial materials for an improved efficiency [23]. The first and most crucial step in improving the sanitation of the processing equipment might significantly influence the quality of the end products. A wide range of sanitizers, such as surfactants, enzymes, and alkali compounds are used in dairy processing industries for eliminating contaminants, by reducing the surface tension, emulsifying fats, and denaturing proteins [19,21]. For instance, Toté et al. observed that chemical disinfectants, such as sodium hypochlorite, hydrogen peroxide, peracetic acid, and isopropanol, could successfully inhibit the bacterial biofilm formation and reduce the viable cells of *Staphylococcus aureus* and *Pseudomonas aeruginosa* in food contact surfaces [24]. Moreover, cell-free supernatants from probiotic bacteria (*Lactobacillus sakei* D.7 and *Lactobacillus plantarum* I.60) have exhibited the excellent eradication efficacy of *L. monocytogenes* biofilm formed in whole milk [10]. To date, several chemical and biologically derived antimicrobial agents are extensively used in dairy processing industries, as summarized in Table 1. Cleaning by chemically derived sanitizers could not remove the surface-associated microorganisms completely and leave the bacterial residual biomass on the dairy processing surfaces which contributes to bacterial regrowth and new biofilm formations [10,20]. Therefore, the selection of appropriate chemicals, acceptable doses, and the proper order of cleaning steps could be used as an effective daily cleaning process for avoiding bacterial contaminations.

### 2.2. Biofilm-Associated Problems in the Meat Processing Industry

The adhesion and formation of mature biofilms by foodborne microorganisms during the manufacturing and handling of fresh meats, have remained a serious concern for consumer health and food safety. Bacterial populations can contaminate non-adulterated carcasses and fresh meat products by spreading through aerosols or direct contact with the surface of slaughter- and manufacturing-related equipment. In the meat processing plant, numerous species of bacteria, including *Escherichia*, *Salmonella*, *Staphylococcus*, *Bacillus*, and *Pseudomonas* spp. could take place to form pathogenic biofilms that primarily contribute to the spoilage of meat products and food-associated infections in the consumers [35]. For instance, beef carcass contamination with *E. coli* O157: H7 may occur while being slaughtered, dressed, chilled, and/or trimmed in the beef processing plant, at a wide range of temperatures [36]. Pathogenic microorganisms have the potential to attach to meat and meat processing surfaces and the expressed specific virulence factors, including adhesins, flagella, curli, fimbria, and enterocyte locus, which play vital roles to initiate and form pathogenic biofilms. Habimana et al. have reported that *E. coli* could be influenced by *Acinetobacter calcoaceticus* and form mixed-bacterial biofilms in the meat processing plant [37]. Hathroubi et al. revealed that surface polysaccharide poly-N-acetyl glucosamine (PGA) could influence the pathogens, such as *A. pleuropneumoniae*, *E. coli*, and *S. aureus* for the antibiotic tolerance and formation of biofilms on meat products [38]. The sheep–goat chain plays an important and significant role in the socioeconomic development of certain countries, mainly in poor and semi-arid zones [35]. Among other probable foodborne pathogens, a staphylococcal contamination was widely reported in bulk goat milk [39]. In sheep and goat meat processing plants, *S. aureus*, coagulase-negative staphylococci (CoNS), *Bordetella parapertussis*, *Bacillus* spp., *Histophilus somni*, and *Pasteurella multocida* were identified as the major pathogenic biofilm formers [40].

Poultry products, such as poultry meats and eggs, are considered as an enriched source of nutrients (e.g., protein) with less fat and have become popular with consumers due to their availability and cheaper prices worldwide [41]. However, poultry products could be contaminated by various foodborne microorganisms, principally, by *Salmonella* and *Campylobacter* spp. [42]. According to the Centers for Disease Control and Prevention (CDC), the pathogenic contamination of boiler meats and eggs could be initiated from numerous sources, including the drinking water supply system on farms and poultry feeds, and causing about 96 million cases of foodborne gastroenteritis illnesses each year, globally [43]. Gazal et al. isolated and identified 117 strains of *E. coli,* after an investigation of commercial chicken processing plants [44]. Among the isolates, 66% of the strains were extended-spectrum *β*-lactamase and AmpC-like enzyme producers, which can effectively degrade the *β*-lactam class of antibiotics (e.g., monobactams and cephalosporins). The frequency of poultry product contamination may rise, due to inadequate knowledge about poultry slaughtering, faulty cutting, and insufficient hygiene practices during production and processing. For instance, *Listeria* spp. was identified from broiler wing meat samples collected from the local market in Hatay province in Turkey [45]. Heidemann et al. have reported about the pathogenic microbial-associated infection pododermatitis in chicken farms and identified 106 bacterial isolates, including *E. coli*, *S. aureus*, *Staphylococcus hyicus*, *Enterococcus faecalis*, *Aerococcus urinaeequi*, *Gallibacterium anatis*, and *Trueperella pyogenes* from the table egg layers [46]. Moreover, using rubber fingers for removing the feathers from carcasses could be considered a potential source of product contamination by pathogenic microorganisms in commercial poultry processing plants [42].

#### Biofilm Control Strategies in the Meat Processing Industry

The existing sanitization approaches are primarily focused on the use of chemical disinfectants, but researchers have already shown that most conventional sanitizers could not completely remove mature biofilm on surfaces in contact with foodstuffs. Treatments using individual sanitizer products demonstrate generally limited efficacy on biofilms, even with extended-time treatment [35,47]. The meat industry thus requires productive and successful products for sanitizing, along with realistic and inexpensive practices to regulate, eliminate, and eradicate biofilms. Through the synergistic effect of biofilm control, new approaches have been drawn up and evaluated, including treatments that use multiple sanitizing agents or combine sanitizers and other cleaning methods. For instance, it has demonstrated a better control and removal of *P. aeruginosa* biofilms by mixing sodium-hypochlorite and hydrogen peroxide, in contrast to each sanitizer individually applied in the same concentrations [48]. Moreover, the displayed sanitizing treatment in conjunction with steam heating with a reduced sanitizer concentration and duration of exposure has increased the rates of suppression of biofilm cells of *E. coli* O157: H7, *S.* Typhimurium, and *L. monocytogenes* [49]. However, the massive use of sanitizers or antibiotics and the rising resistance of pathogens are growing concerns. Therefore, natural antibacterial compounds came forward as alternatives to traditional therapeutics. Antimicrobial and bioactive molecules (e.g., bacteriocin, bacteriocin-like inhibitory substance [BLIS], enzyme) from beneficiary microorganisms and plants, as a source, have contributed to the development of new drugs to combat many diseases and biofilm inhibition by battling the bacterial QS mechanism, suppressing genes, and decreasing the development of the surface structure [50]. For instance, phloretin, a dihydrochalcone flavonoid present in apples, which have the potential to suppress the genes responsible for producing QS molecules, toxins, prophages, and curli development, thus inhibiting the biofilm formation by *E. coli* O157: H7 in meat [51,52]. In the porcine meat industry, antimicrobial treatments are therefore threatened in *A. pleuropneumoniae* with an antibiotic resistance and biofilm formation. Thus, the chemical inhibitor Phe-Arg-β-naphthylamide (PAßN) can suppress the establishment of biofilm and eliminated the mature biofilm formed by *A. pleuropneumoniae*. It has been exhibited that the use of PaβN, combined with ceftazidime or ofloxacin could inhibit the *A. pleuropneumoniae* biofilm formation more efficiently than PAβN alone in pig meat [53]. Additionally, practicing good hygiene and using proper disinfectants (e.g., iodine, chlorine, quaternary ammonium, and chlorhexidine) have also exhibited a high efficacy against *C. pseudotuberculosis*-associated biofilm prevention (84.4–100%) in sheep and goat farm [35].

The eradication of *Salmonella* biofilm from poultry is well known to be troublesome. For the removal of *Salmonella*, various chemical options are required for commercial purposes. However, different trials in poultry environments found that after removing and sanitizing *Salmonella* in broilers and hen houses, there was a high incidence of decontamination toward bacteria at the field level [35,41]. Biofilm can be eliminated by integrating treatments with different spectrums and mechanisms of action, which involve chemical, natural, and physical treatments, and have been assessed for that purpose. The application of physical approaches, such as high- and low-intensity ultrasounds, could be used as a green and promising emerging technique for inhibiting the formation of pathogenic biofilms in poultry products. For instance, using the ultrasound treatment (45 kHz and 1.6 W/cm^2^) during the processing could enhance the quality of chicken breast meat and Korean soup (Baeksuk) made from chicken broth [48]. However, using the combination of ultrasound with slightly acid-electrolyzed water during the pre-chilling of chicken carcasses could enhance the reduction of the pathogenic microorganisms from chicken breast meat [54]. Moreover, different biological agents and natural compounds derived from beneficiary microorganisms and plants could also exhibit an antibiofilm efficacy against a broad-spectrum of pathogenic microorganisms. For instance, plant-derived phenolic compound quercetin could downregulate the virulence, stress response, and QS genes and inhibit the biofilm formation of *Salmonella* spp. on chicken skin and poultry-processing surfaces [41]. Recently assessed, the antibacterial susceptibility and biofilm elimination by nalidixic acid, combined with natural compounds (e.g., eugenol, thymol, and carvacrol) had exhibited an effective antibiofilm efficacy against twelve strains of *S.* Typhimurium in poultry processing plants [55]. A list of natural compounds inhibiting the pathogenic microbial biofilms is summarized in Table 2.

### 2.3. Biofilm Associated Problems in the Aquatic Industry

Fishing is one of the oldest activities carried out by humans, dating back to prehistoric times and aquacultures have a high demand due to their important role in the world economy, particularly in coastal communities and developing countries. To date, approximately 0.6 billion people (10% of the total world population) primarily rely on the aquatic biodiversity for their livelihood and subsistence (Food and Agriculture Organization [FAO]) [65]. From a world hunger and nutrition standpoint, aquatic foods are considered as the major protein source and could be the best alternative to animal-derived proteins [66]. Aquatic foods can be contaminated and decay rapidly during any stage of the production and distribution process, due to the biochemical degradation and the presence of pathogenic bacteria on their surfaces after capture. Zoonotic bacterial species, such as *Staphylococcus* spp., *Pseudomonas* spp., *Listeria* spp., *Salmonella* spp., *Vibrio* spp., *Aeromonas hydrophila*, and *E. coli* are primarily responsible for the biofilm formation and aquatic-associated disease outbreaks worldwide [3]. The failure of proper handling and inadequate sanitation procedures in aquatic processing facilities is deemed to be the persistence of aquatic-associated bacteria in aquatic foods and food contact surfaces [67]. Fresh and salt water fishes become colonized with pathogens in their surfaces or inner organs from polluted aquatic environments, that lead to the formation of bacterial biofilms [68]. For instance, *S. aureus* is repeatedly detected in fishery products, which is responsible for the foodborne intoxications (e.g., staphylococcal enterotoxins [SEs]) in humans worldwide. To date, a total of 23 SEs-associated genes (e.g., *sea*, *seb*, *sec*, *sed*, *see*, *seg*, *she*, *sei*, *selj*, *sek*, *sel*, *sem*, *sen*, *seo*, *seq*, *sep*, *ser*, *ses*, *set*, *selu*, *selu2*, *selv*, and *selx1*) were reported after the screening of 1545 *Staphylococcus* spp. and 97% of *S. aureus* having one or more enterotoxigenic (ET) genes in their genome [69]. Ham et al. identified the presence of *Staphylococcus* spp. in 33.8% dried seasoned fish products, among 210 samples, which were collected from the South Korean retail market [70]. A study conducted by Moon et al. exhibited that ET gene *se-*carrying *S. aureus* could cross-contaminate aquatic food products and facilitate the biofilm formation under refrigerated conditions [71]. The thermal- and protease-resistance nature of ET produced by *Staphylococcus* spp. could retain their emetic activity even after marine food processing and enhance the risk of intoxication [20]. Several studies have also reported the persistence of *Leptospira* spp., *Yersinia* spp., *L. monocytogenes*, *Aeromonas hydrophila*, and *Francisella tularensis* in aquatic food processing facilities, due to inadequate handling and ineffective sanitizing procedures [67]. *L. monocytogenes* serotypes 1/2a, 1/2b, 1/2c, and 4b are frequently found in both fisher products and fish-processing contact surfaces and are considered the causative agent of human listeriosis (e.g., febrile gastroenteritis and systemic infections). Skowron et al. identified 237 *L. monocytogenes* isolates after investigating the fish products and fish-processing surfaces and found a total number of 161 genetically dissimilar strains, via the pulsed-field gene electrophoresis method [72].

#### Biofilm Control Strategies in the Aquatic Industry

Aquatic foods (e.g., fish and fishery products) are considered one of the major export and import food commodities in the world trade market [3]. Therefore, retaining the aquatic food quality, safety, and processing standards might promote product acceptance by consumers and traders in the international market [65]. Conventionally different physical and chemical approaches, including salting, drying, chlorination, and ultraviolet (UV) treatments, are used as antimicrobial agents in the aquatic food industry from the early period of food processing [73]. To date, advanced technologies are used to enhance the efficacy of traditional approaches against aquatic food-associated microorganisms. For instance, the treatment by light-emitting diodes (LEDs) can inhibit the biofilm formation by reducing the pathogenic microbial populations of *Vibrio parahaemolyticus* and *S. aureus* on aquatic foods [74]. Fan et al. reported the combined physical treatment of UV-C with LEDs against foodborne pathogens, such as *S.* Typhimurium, *E. coli* O157: H7, and *L. monocytogenes* on raw tuna [75]. A physical approach, such as ultrasound, can significantly enhance the bactericidal activity against both planktonic and biofilm forms. For instance, using high-power ultrasound (25 kHz), could inhibit the microbial biofilm formation and enhance the quality of codfish fillets [76]. Ovissipour et al. reported that the efficacy of using electrolyzed waters (e.g., natural and acidic) as antibiofilm agents, could be influenced by the rising temperature and effectively reduce the bacterial populations of *L. monocytogenes* in Atlantic salmon, without changing the fish proteins [77]. Moreover, acidic electrolyzed ice water could enhance the quality of shrimp by limiting the pH change in shrimp flesh and inhibiting the microbial growth on raw shrimp surfaces [78]. Another study revealed that smoked salmon fillets treated with nonthermal dielectric barrier discharge plasma for 60 min, reduced about 90% of *L. monocytogenes* [79]. However, several biological agents (e.g., bacteriocins, BLIS, and enzymes) are extensively used as antibiofilm agents in the aquatic food industry [68]. For instance, seafood treated with bio preservatives, such as bacteriocin and essential oils could efficiently inhibit the planktonic and biofilm cells of twelve *L. monocytogenes* strains [80]. For instance, the postbiotic components derived from *Leuconostoc mesenteroides* J.27 and essential oils (e.g., eugenol and thymol) successfully inhibited the pathogenic microorganisms, such as *Vibrio parahaemolyticus*, *P. aeruginosa*, and *E. coli* O157:H7 [3]. Moreover, the inhibition rate of each pathogen from seafood (*Todarodes pacificus*) and food-processing surfaces was increased after a combination treatment of postbiotics with essential oils. However, advanced physical or chemical approaches, combined with new technologies, such as the inhibition of QS molecules, enzymatic disruption, bactericidal coating, nanotechnology, and bioelectric approach, have successfully been studied to find effective alternatives for the prevention and control of biofilms [81]. The combination of two or more antimicrobial approaches, called the “hurdle” technique, could be used in the aquatic food industry as an alternative to the conventional approach. For instance, the combination of acidic electrolyzed oxidizing water with lysozyme enzyme could enhance the quality and storage time by inhibiting bacterial biofilm formation on carp fish [82]. Therefore, hurdle technology is gaining more attractiveness and acceptance due to its potentiality, including eco-friendly, cost-effective, and the lack of deterioration of food contact surfaces.

### 2.4. Biofilm Associated Problems in the Agricultural Industry

Plant-microbe interactions have a necessary influence on plant nutrition, growth, biocontrol, and stress alleviation. The equilibrium of soil nutrients is also dependent on the interactions via physical, chemical, and biological properties persuaded by biogeochemical cycles in the soil [83]. The presence of pathogenic bacteria in the environment (e.g., soil and water) might adhere and colonize plant surfaces during pre-harvesting (propagation) and post-harvesting (processing), which can lead to biofilm-associated problems in the agricultural industry. The colonization of pathogenic microorganisms could occur on plants via seeds, roots, leaves, stems, and vascular tissues (i.e., xylem and phloem). Following the adhesion of pathogenic microorganisms on plant tissue surfaces, the microcolony formation of pathogenic cells turns into massive biofilm structures via the plant-microbe interaction (e.g., pathogenesis, mutualism, or commensalism) [84]. The nutrient and water accessibility in plant tissue surfaces, in particular the proclivity of the microbial colonization, thus manipulates the formation of pathogenic cell clusters in biofilms [7]. Bacteria-associated microbial hazards are mostly observed in freshly produced agriculture products and are responsible for foodborne diseases worldwide [85,86]. To date, many outbreaks have been associated with the consumption of freshly produced agriculture products, including carrots, lettuce, cucumbers, onions, spinach, and tomatoes due to the surface colonization by the biofilm-forming pathogens (e.g., *Salmonella* spp., *Campylobacter* spp., *Vibrio* spp., *Shigella* spp., *Clostridium* spp., *L. monocytogenes*, *E. coli*, *Aeromonas hydrophila*, and *Bacillus cereus*) [83,87]. The use of contaminated soil and water for plant irrigation could act as a reservoir and route of pathogenic microbial transmission that causes foodborne illness to consumers [13]. The diversity of multicellular assemblies of microbes on plant surfaces varied, in terms of morphology from microcolony formations, aggregates, and clusters in specific or scattered locations. Additionally, numerous factors (e.g., age of biofilms, nutrient levels, oxygen levels, EPS, aggregation, waste product accumulation, mechanical signals, host-derived signals, antimicrobials, biocides, metal ion concentrations, and plant volatiles) have significant roles during the plant-associated biofilm formation [84]. Recently, the CDC has reported on the increments of fresh produce-associated disease outbreaks worldwide in the last decade compared to other food products [43]. The investigation carried out on fresh produce-associated human illness in European countries reported that *Salmonella* spp. had the highest presence (0.1–2.3%) in fresh-cut fruits and vegetables [88]. The European Food Safety Authority (EFSA) reported that more than 10% of outbreaks were linked with freshly produced food products from 2007 to 2011, which accounted for the hospitalization of approximately 35% of people and 46% of deaths in Europe [89]. Recently, *Salmonella* spp. has been reported as the major zoonotic bacteria found in fresh papayas and pre-cut melons, which caused about 188 cases in different states of the USA, in 2019 [87]. However, fresh-produce-associated outbreaks are mostly reported in developed countries (e.g., Canada, USA, Australia, and the European continent), compared to developing or underdeveloped countries, due to their insufficient technology for the surveillance of foodborne-associated diseases.

#### Biofilm Control Strategies in the Agricultural Industry

Biofilm formations by pathogenic microorganisms in fresh produce can not only protect the pathogenic cells from common antimicrobial agents but can also fight against the immune system of humans [13]. Freshly produced foods (e.g., fruits and vegetables) are mostly treated non-thermally, during their processing procedures. Therefore, the selection of potential antimicrobial agents is crucial for enhancing product quality and shelf-life by reducing the perishability of processed fruits and vegetables [86]. For avoiding the microbial biofilm formation on agricultural food products, we consider that the preventive approach could be preferable to treating the existing microbial biofilms [82]. For instance, good manufacturing and hygiene practices, critical control points, and a proper hazard analysis in agricultural food processing facilities can prevent the pathogenic microbial adhesion and the following biofilm formation on freshly produced foods and food processing surface materials [83]. Several conventional approaches have frequently been used from the earliest time to minimize foodborne microbial contamination during the pre- and post-harvesting of agricultural products. For instance, considering the physical countermeasures, including the hypobaric storage, low storage temperature, and cold atmosphere storage could use as a preventive or remedial antimicrobial agent, during the processing period of fruits and vegetables [85]. Moreover, preventive approaches, such as prior to the harvesting of fruits and vegetables in farms, following the go-slow rules of about 90 to 120 days for all forms of soil alteration and irrigation after composting, could inhibit bacterial infection in fresh products [87]. For instance, the application of a thermal approach during composting can reduce the *Clostridium difficile* and *Clostridium perfringens* associated problems in agricultural farms [90]. The stoppage of water irrigation for about 1 to 10 days in agriculture farms before harvesting can effectively reduce the bacterial infection of *E. coli* O157: H7, *L. monocytogenes*, and *Salmonella* spp. in fresh produce [91]. Food irradiation technology (e.g., ionizing radiation) could effectively control the foodborne microorganisms during the postharvest period of freshly-cut produce. For instance, ionizing radiation treatment (2.5–3.0 kGy) applied on the post-drying apple could minimize the pathogenic microbial-associated hazard and enhance the shelf life of apple chips without changing the food texture and color in the storage period [92]. Additionally, several biological agents (e.g., enzymes, probiotics, bacteriophages) have also the potential to inhibit the foodborne microorganisms in fresh products [4,7]. Hossain et al. reported that the probiotic potential lactic acid bacteria could effectively inhibit the human listeriosis-occurring bacteria *L. monocytogenes* in freshly-cut lettuce and food processing surfaces [93]. However, the combination of commercial bacteriocin (nisin) with food-grade essential oils (eugenol and thymol) exhibited strong antibiofilm efficacy against *L. monocytogenes* on lettuce and food-processing surfaces [4]. The chemical-based cheap reconditioning agent chlorine is frequently used as a disinfectant in the agricultural industry due to its availability, economic efficiency, minor impact on food quality, and effectiveness against both enteric and vegetative bacteria, as well as viruses. Van Haute et al. reported that spraying or washing with free chlorine (1 mg/L) could effectively decrease the *S. enterica* serovar Thompson and *E. coli* O157: H7 bacteria population from the butter-head lettuce (*Lactuca sativa*) [94]. Moreover, the plant-microbe interaction and biofilm formation by agriculture-friendly microorganisms could have beneficiary effects in the agro-ecosystem, which may enhance the productivity of fresh products by inhibiting the pathogenic microorganisms [84]. A list of plant-beneficial microorganisms and their role in agriculture is summarized in Table 3. However, viable but nonculturable (VBNC) pathogens from the environment are found to be resistant to conventional antimicrobial agents and the dormant bacterial cells could persist during the harvesting period, leading to the formation of foodborne pathogenic biofilms on fruits and vegetable products. Therefore, the execution of rapid, sensitive, and specific detection and quantification methods, such as propidium monoazide-quantitative PCR combined with a loop-mediated isothermal amplification assay could effectively identify and reduce the hazard of VBNC bacteria (e.g., *E. coli* O157: H7 and *S. enteric*) in fresh produce [95].

## 3. Current Challenges and Prospects

The elegant and sophisticated characteristics of the microbial communities and their involvement in the devastation of the food industry make them interesting to study and have encouraged the development of a complete scenario from the biofilm formation to their remediation actions, in the last two decades. However, it is not easy to control and eradicate the biofilms formed by pathogenic microorganisms in foods and food contact surfaces, due to the potential scopes of microbial cross-contamination during the food processing of raw products, production, storage, and delivery. Zwirzitz et al. reported after an investigation of a pork-processing plant about the sources of bacterial contamination and the presence of environmental pathogens in pork meat, which were largely non-meat-associated foodborne microorganisms [47]. Following the inspection of the meat conveyor belts, different foodborne pathogens were found to be present, even after daily cleaning and disinfection procedures of the meat-processing plant [102]. Therefore, finding effective measures to prevent microbial biofilm formation or to eradicate mature biofilms from foods and food contact surfaces remains a big challenge. Due to the existence of miscellaneous matrix components in microbial biofilms, it is difficult to break down the mature biofilm and eradicate the microbial cells with a single-strength antimicrobial approach. Therefore, the combination of two or more antimicrobial approaches could overcome this scantiness.

Additionally, more and more questions have been raised about the extensive use of the chemicals leading to significant issues, including surface area degradation, the atmospheric transmission of leftover antimicrobial agents, or a rise in cross-resistance to antibiotic sanitizers amongst other items. Therefore, innovative alternatives were suggested to be used in the food industry, as cleaning agents and sanitizers or to supplement existing methods for sanitization [34]. For instance, bio-enzymes *α*-amylase or *β*-glucanase combined with antimicrobial surfactant cetyltrimethylammonium bromide could increase the eradication of *P. fluorescens* biofilms and inhibit the bacterial regrowth on food contact surfaces (e.g., stainless-steel) [103]. Additionally, it has been proved that the combination of sodium hypochlorite with enzymes cellulase and proteinase K could effectively reduce the bacterial biofilms (17–37%) formed by *E. coli* O157: H7 on a polystyrene surface [104].

The implementation of advanced bacterial biofilm prevention and control approaches is a comprehensive strategy that takes into consideration the microbiological standard of inputs, the purifying and decontamination treatments, the atmospheric conditions of the site, the physical and chemical properties, and the proper sanitary layout of materials and utensils of the bacterial contaminants. It is obvious from this point of view that more feasible washing and decontamination treatments are needed for the rapid control of pathogens. In designing food processing equipment and coatings, it is necessary to take into account the convenience of their wash and availability for cleanliness processes. It is necessary to expand the understanding of the cellular mechanisms behind the development and behavior of microbial biofilms for controlling the microbial contamination and inhibiting their biofilm formation in food processing plants. Moreover, broad studies are needed about the different chemical, biological, and physical antibiofilm approaches before their applications in food processing industries. A greater understanding is also necessary about (i) the factors related to the biotic and the abiotic surfaces in bacterial colonization, (ii) the sessile growth control system and metabolic processes, (iii) within these biofilm populations the connection between bacteria, is critical in the enhancement and implementation of sanitation policies and procedures that can efficiently remove and avoid product contamination, due to spoilage and pathogens, taking into account the multifaceted aspect of microbial biofilm formation. However, all emerging antibiofilm approaches should be tested in in vitro and in vivo studies for verification and to avoid health and environmental hazards after their application.

## Figures and Tables

**Figure 1 microorganisms-10-02348-f001:**
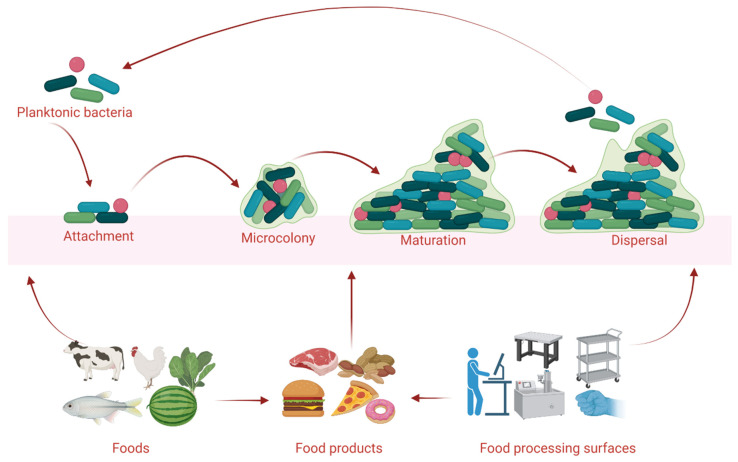
The formation and development of bacterial biofilms in foods and food processing surfaces in agri-farm industries. The figure was illustrated using BioRender.com (accessed on 22 July 2022).

**Table 1 microorganisms-10-02348-t001:** Antimicrobial products used for controlling bacterial biofilms in dairy processing facilities.

Antimicrobial Products	Target Bacteria	References
Peracetic acid	*L. monocytogenes*	[25]
Ozone	*P. fluorescens* and *P. aeruginosa*	[26]
Sodium Hypochlorite	*S. aureus*	[24]
Linoleic acid	*K. pneumonia*	[27]
Quercetin	*P. aeruginosa*, *K. pneumonia*	[28]
Fisetin	*S. aureus*	[29]
Chitosan	*Staphylococcus* spp.	[30]
Ellagic acid	*S. aureus*	[31]
Hydrogen peroxide; sodium dichloroisocyanurate	*S. aureus*	[32]
Sodium hydroxide	*P. putida*	[33]
Chlorinated-alkaline solution; low-phosphate buffer detergent; alkaline solution; hypochlorite	*L. monocytogenes*	[34]

**Table 2 microorganisms-10-02348-t002:** List of natural compounds inhibiting the pathogenic microbial biofilms.

Natural Antibiofilm Compound	Source	Target Species	Relative Inhibitory Expression on Virulence Genes	References
Polyphenol epigallocatechin gallate	Green tea (*Camellia sinesis*)	*Stenotrophomonas maltophilia*, *E. coli*, *S. aureus*, *E. faecalis*, *P. aeruginosa*, and *Streptococcus mutans*	Reducing the expression of biofilm regulatory gene *CsgD*	[56]
Indole-3-acetaldehyde	*Rhodococcus* sp. strain BFI 332	*E. coli* and *Candida tropicalis*	Curli operons, *csgBAC*, and *csgDEFG*	[57]
Cinnamon bark oil, cinnamaldehyde	Plant	*P. aeruginosa*, *E. coli*, *Porphyromonasgingivalis*, *L. monocytogenes*, and *Staphylococcus epidermidis*	Curli and Shiga toxin genes	[58]
Resveratrol oligomers	Plant	*E. coli*, *P. aeruginosa*	Fimbriae inhibition	[59]
Trans-resveratrol and its dimer, ε-viniferin	Extract of Carexdimorpholepis	*E. coli*, *P. aeruginosa*	Repression of curli and swarming motility genes	[60]
Honey	Honeybee	*E. coli*, *P. aeruginosa*	Curli genes (*csgBAC*) and virulence genes	[61]
Essential oil (eugenol, thymol)	Bay, clove, and pimento berry	*E. coli*, *Pseudomonas* spp., *S. aureus*	Type 1 fimbriae genes, curli forming gene	[62]
5-iodoindole, 7-hydroxyindole	*Acinetobacter calcoaceticus*	*Acinetobacter baumannii*, *E. coli*, *P. aeruginosa*, *S. aureus*, and *Candida albicans*	Induces the biofilm inhibitor regulator *ycfR* and inhibits motility genes	[63]
Exopolysaccharides	*Lactobacillus acidophilus* A4	*E. coli*, *Salmonella* spp., and *P. aeruginosa*	Curli production (*crl*, *csgA*, and *csgB*) and chemotaxis (*cheY*)	[50]
Bergamottin, dihydroxybegamottin	grapefruit juice	*E. coli, Salmonella* spp., *S. aureus*	AI-2 related genes	[64]

**Table 3 microorganisms-10-02348-t003:** List of the beneficiary plant pathogens and their roles in agriculture.

Plant Pathogens	Genes	Function	References
*Agrobacterium* *tumefaciens*	*Exo*, *chvA*, *chvB*, *celD*, *celE*, *phoB*	Cellulose synthesis; production of EPS, cell attachment, virulence, and biofilm formation	[96]
*Aspergillus* *nidulans*	*MsbA*	Regulates nitrogen-activated protein, kinase signaling, and biofilm formation	[97]
*Pantoea agglomerans*	*purB*	Rhizosphere colonization	[98]
*Fusarium oxysporum* f. sp. *cucumerinum*	*FocVel1*	Velvet genes involved in biofilm formation	[99]
*Xanthomonas axonopodis*	*Exo*	EPS biosynthesis and virulence	[83]
*Xanthomonas citri* subsp. *citri*	*gumB*, *gumD*, *galU*	Xanthan production	[100]
*X*. *citri* subsp. *citri*	*hrpM*, *gumB*, *gumD*, *galU*, *hrpM*, *sahH*	Periplasmic glucan biosynthesis, biofilm formation	[101]

## Data Availability

Data is contained within the article.

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
