# Peer review of "Pernicious Attitude of Microbial Biofilms in Agri-Farm Industries: Acquisitions and Challenges of Existing Antibiofilm Approaches"

_microorganisms, 2022, doi:10.3390/microorganisms10122348_

Round 1

Reviewer 1 Report

Manuscript details:
Journal: Microorganisms
Manuscript ID: microorganisms-2056472
Type of manuscript: Review
Title: An ugly attitude of microbial biofilms in agri-farm industries: acquisitions and challenges of existing antibiofilm approaches

In the manuscript entitled “An ugly attitude of microbial biofilms in agri-farm industries: acquisitions and challenges of existing antibiofilm approaches”- summarized the present microbial biofilm-associated problems in agriculture sectors. The authors explained briefly several effective ways to minimize those problems. They also provided the limitations of different remedial techniques and provided probable solutions to overcome them. However, a few comments that should be addressed before this manuscript could be considered for publication are provided below. In addition, I would like to minor revisions before publication. Below are the suggestions to improve the manuscript. 

  1. L21: antimicrobial
  2. L21, 27: antibiofilm
  3. L33: Authors are suggested to follow the journal reference style.
  4. L43: Authors are suggested to cite the most updated reference (last 5 years) in the introduction section.
  5. L72, 73, and throughout the manuscript: antimicrobial
  6. L72: include “10-1000 times more resistant”
  7. L89: Replace “food contact surfaces” with “food-processing surfaces”
  8. L159: S. aureus
  9. L208: Use the full form of any abbreviated words first time throughout the manuscript.
  10. L269: 1,545
  11. L291: put space “to consumers”
  12. L480: investigation
  13. Many spacing and punctuation marks problems are found throughout the manuscript. Revision required.

Reviewer 2 Report

The manuscript presents the pathogenic biofilm-related problems in agri-farm industries, potential measures, and finding suitable ways to address the challenges of current anti-biofilm approaches. I would like to request addressing the following issues before this manuscript can be accepted for publication.

Major issues.

There are few literatures in recent years. As a review, it is suggested to add some recent research results

Minor issues.

Line 46, revise this sentence.

Line 48, Uniform format, "lipids"?

Line 315, Here the abbreviation of "quorum sensing" should appear when it first appears on line 211.

Round 2

Reviewer 2 Report

  • A good modification has been made according to the previous comments.
  •